# *Gcd* Gene Diversity of Quinoprotein Glucose Dehydrogenase in the Sediment of Sancha Lake and Its Response to the Environment

**DOI:** 10.3390/ijerph16010001

**Published:** 2018-12-20

**Authors:** Yong Li, Jianqiang Zhang, Zhiliang Gong, Wenlai Xu, Zishen Mou

**Affiliations:** 1Faculty of Geosciences and Environmental Engineering, Southwest Jiaotong University, Chengdu 610059, China; lyswjtu@sohu.com; 2School of Food and Biological Engineering, Xihua University, Chengdu 610039, China; 0120020092@mail.xhu.edu.cn; 3State Key Laboratory of Geohazard Prevention and Geoenvironment Protection, Chengdu University of Technology, Chengdu 610059, China; mouzishen17@cdut.edu.cn

**Keywords:** *gcd* gene, diversity, eutrophication, Sancha Lake, environmental factors

## Abstract

Quinoprotein glucose dehydrogenase (GDH) is the most important enzyme of inorganic phosphorus-dissolving metabolism, catalyzing the oxidation of glucose to gluconic acid. The insoluble phosphate in the sediment is converted into soluble phosphate, facilitating mass reproduction of algae. Therefore, studying the diversity of *gcd* genes which encode GDH is beneficial to reveal the microbial group that has a significant influence on the eutrophication of water. Taking the eutrophic Sancha Lake sediments as the research object, we acquired samples from six sites in the spring and autumn. A total of 219,778 high-quality sequences were obtained by DNA extraction of microbial groups in sediments, PCR amplification of the *gcd* gene, and high-throughput sequencing. Six phyla, nine classes, 15 orders, 29 families, 46 genera, and 610 operational taxonomic units (OTUs) were determined, suggesting the high genetic diversity of *gcd*. *Gcd* genes came mainly from the genera of *Rhizobium* (1.63–77.99%), *Ensifer* (0.13–56.95%), *Shinella* (0.32–25.49%), and *Sinorhizobium* (0.16–11.88%) in the phylum of Proteobacteria (25.10–98.85%). The abundance of these dominant *gcd*-harboring bacteria was higher in the spring than in autumn, suggesting that they have an important effect on the eutrophication of the Sancha Lake. The alpha and beta diversity of *gcd* genes presented spatial and temporal differences due to different sampling site types and sampling seasons. Pearson correlation analysis and canonical correlation analysis (CCA) showed that the diversity and abundance of *gcd* genes were significantly correlated with environmental factors such as dissolved oxygen (DO), phosphorus hydrochloride (HCl–P), and dissolved total phosphorus (DTP). OTU composition was significantly correlated with DO, total organic carbon (TOC), and DTP. GDH encoded by *gcd* genes transformed insoluble phosphate into dissolved phosphate, resulting in the eutrophication of Sancha Lake. The results suggest that *gcd* genes encoding GDH may play an important role in lake eutrophication.

## 1. Introduction

In the process of growth and reproduction of phosphate-solubilizing microorganisms in water environments, some organic acids are produced and secreted, reducing the environmental pH, transforming insoluble phosphate into dissolved phosphate, promoting the growth and development of cyanobacteria, and causing eutrophication of water [1]. Among the dissolving phosphoric acids produced by microorganisms, gluconic acid (GA) is the most important and main one [2]. Its metabolism is strictly regulated by enzymes in microorganisms. Quinoprotein glucose dehydrogenase (EC1.1.5.2, GDH) is the most fundamental and key enzyme in microbial phosphorus metabolism. GDH catalyzes the oxidation of glucose into GA with pyrroloquinoline quinone (PQQ) as the prosthetic group [3]. At the moment, studies of GDH focus on gene characteristics and functions by purification culture. The *gcd* gene encoding GDH has been obtained in domestic and foreign studies. Goldstein conducted the *gcd* gene expression of *Erwinia herbicola* in *Escherichia coli* HB101, and the final metabolite GA showed the ability to dissolve mineral phosphorus [4]. Cleton-Jansen et al. cloned the *gcd* gene from *E. coli*, which produced GDH enzymes, and they mapped and sequenced the cloned *gcd* gene [5]. Tripura et al. cloned the *gcd* gene encoding GDH from *Enterobacter asburiae*, and the bacteria could not dissolve insoluble phosphate after the gene mutation [6]. The GDH *gcd* gene (pg5SD2) was cloned from the *Pseudomonas frederiksbergensis* strain by Qingwei Zeng et al., and it was found that GDH, a product of pg5SD2 translation, can regulate mineral phosphate solubilization (MPS) [7]. These studies of *gcd* by purification culture cannot represent the composition and diversity of *gcd*-harboring bacterial communities objectively.

At present, few studies have been conducted on the gene diversity of culture-free bacteria in the natural environment. From soil samples of the Dianchi Lake drainage area of China, Peixiang Yang et al. screened 123 strains of *gcd*-harboring bacteria which belong to three bacterial phyla and 12 genera [8]. By referring to the literature related to *gcd* design primers and amplifying *gcd*, Fabian Bergkemper et al. obtained six phyla, 17 orders, and 24 families of *gcd*-harboring bacteria from German forest soil samples. There were abundant and diverse *gcd*-harboring bacteria in the soil environment that play an important role in the metabolism of MPS [9]. None of these studies did covered *gcd* gene diversity. No research has been conducted on the diversity of *gcd* genes and its relationship with eutrophication in lake sediments. For this study, our aims were to define which *gcd* genetic backgrounds were generated in lake sediments, and what factors were related to *gcd* genetic distribution, to reveal the relationship between lake eutrophication and *gcd*-harboring bacterial communities. This paper studied the diversity as well as the spatial and temporal distribution of *gcd* in Sancha Lake sediments and its response to environmental factors such as dissolved oxygen (DO), total organic carbon (TOC), total phosphorus (TP), and dissolved total phosphorus (DTP) by DNA extraction of sediments, PCR amplification, and high-throughput sequencing in order to explore *gcd* gene diversity and its response to the environment, which is useful to control eutrophication. We hypothesized that *gcd*-harboring bacteria were diverse and varied by season and sampling site type. *Gcd* genes encoding GDH were closely related to lake eutrophication.

## 2. Materials and Methods

### 2.1. Site Description and Sample Collection

Situated in Tianfu New District of Chengdu, Sichuan Province, Sancha Lake is located at 104°11′16″ to 104°17′16″ east longitude and 30°13′08″ to 30°19′56″ north latitude, with an average depth of 8.3 m and a maximum depth of 32.5 m. This region is in the subtropical humid monsoon climate zone, with an average annual temperature of 15.2–16.9 °C and average annual precipitation of 786.5 mm. The water source of Sancha Lake is mainly from the Min River, accounting for about 80% of the total water volume of the reservoir, and about 20% comes from rainfall and two creeks. The area of the basin above the dam site is 161.25 km^2^. Sancha Lake is a vital source of drinking water. According to the monitoring results over years, the inflow of COD_Cr_ and BOD_5_ into Sancha Lake shows a downward trend year by year for the control of external pollutants. However, the total nitrogen (TN) and TP of the water body has been on an upward trend. The corresponding chlorophyll increases on a yearly basis, while the transparency decreases year by year, and the water has been eutrophicated [10]. Sancha Lake eutrophication might be attributed to insoluble phosphate transformed to soluble phosphate by *gcd*-harboring bacterial communities.

According to the characteristics of sediment distribution and eutrophication of Sancha Lake, six sampling sites were selected, as shown in Figure 1. The longitude and latitude of the sampling sites were determined by GPS. The characteristics information of all sampling sites is presented in Table 1. At each sampling site, there were three sampling points. In April (spring) and November (autumn) 2017, a claw-like Peterson dredge was used to capture surface sediments at each sampling point (Figure 2). The surface (0–5 cm) sediment was collected by a plexiglass column and put into a clean, sealed polythene bag [11]. Three parallel samples were collected at each sample point and mixed as the representative sample of the sample point. Some sediments were stored at 4 °C for physical and chemical analysis (within 24 h), and some sediment samples were stored at −80 °C for DNA extraction. At the same time, an air-tight water sampler was used to collect the overlying water above the sediment layer at each sampling point for the water environment index analysis [11].

### 2.2. Characterization of Sediment Physicochemical Properties

Several morphological grading experiments of phosphorus in sediments were performed using the Standards Measurements and Testing Program of the European Commission (SMT) [12,13]. Determination of TP, inorganic phosphorus (IP), organic phosphorus (OP), phosphonium hydroxide (NaOH–P), and HCl–P in sediments was conducted, and the content of different forms of phosphorus was determined by ammonium molybdate spectrophotometry [14]. Total organic carbon in sediments (determination of total organic carbon in sedimentary rocks) was determined by the Chinese method GB/T 19145-2003 [15]. The determination of total nitrogen was determined by alkaline potassium persulfate digestion by UV spectrophotometry [16]. Ammonium nitrogen content was measured with the extraction of potassium chloride solution spectrophotometric method [17]. The determination method of overlying water DTP was as follows: the content of phosphorus was determined by the ammonium molybdate spectrophotometric method [14] after the water was filtered by a 0.45-μm removal filter membrane, the pH value and temperature was measured by the portable multiparameter temperature meter HI991301, and the dissolved oxygen was gauged by the HQ3OD portable dissolved oxygen meter.

### 2.3. DNA Extraction, PCR, and Illumina MiSeq Sequencing

The total DNA was extracted after centrifugation to remove excess water from the sediment. The centrifugal sediment samples were then added to a Power Bead Tube following the steps according to the manufacturer with adjustment only for shock duration. The concentration and purity of DNA was analyzed with NanoDrop2000. The DNA extracts were purified with a Wizard DNA Clean-Up System (Axygen Biosciences, Union City, CA, USA), as recommended by the manufacturer. The DNA was stored at −80 °C until analysis. The GDH-encoding gene *gcd* was amplified by PCR from DNA using *gcd*-FWCGGCGTCATCCGGGSITIYRAYRT and *gcd*-RW GGGCATGTCCATGTCCCAIADRTCRTG [9]. The amplicon size was 330 bp. Each sample involved three technical replicates for *gcd* gene amplification, which was carried out in TaKaRa Taq™ Hot Start Version (New England Biolabs, Ipswich, MA, USA). Indexing PCR was performed in 25-μL reactions containing: 2.5 μL buffer (10×), 2 μL 2.5 mMdNTPs, 2.5 μL forward primer (5 μM), 2.5 μL reverse primer (5 μM), 0.3 μL rTaq polymerase, 10 ng template DNA, and ddH_2_O added to bring to volume. The amplification procedure included an initial denaturation step (95 °C; 12 min), 10 cycles of denaturation (95 °C, 30 s; 65 °C, 1 min; 72 °C, 30 s), 40 cycles of annealing (95 °C, 30 s; 54 °C, 1 min; 72 °C, 30 s), and elongation (72 °C; 5 min).

PCR products were recovered using 2% agarose gel and purified using AxyPrep DNA Gel Extraction Kit (Axygen Biosciences, Union City, CA, USA). Then, Tris-HCl elution and 2% agarose gel electrophoresis were conducted. Quantitative detection was performed by QuantiFluor™-ST (Promega, Madison, Wisconsin, USA). The purified amplified fragments were constructed according to the standard operating procedures of Illumina MiSeq platform (Illumina, San Diego, CA, USA). Shanghai Majorbio Bio-pharm Technology Co., Ltd. (Shanghai, China) was used for the sequencing using an Illumina MiSeq platform PE300.

### 2.4. The Diversity and Composition of gcd-Harboring Bacterial Communities Were Assessed Using High-Throughput Sequencing Technique

The original sequencing sequence used Trimmomatic software for quality control and adopted FLASH (version 2.7, http://ccb.jhu.edu/software/FLASH/, Center for Bioinformatics and Computational Biology, Iowa City, IA, USA) for sequence assembly. Operational taxonomic unit (OTU) clustering was performed at 97% similarity by UPARSE software (version 7.1 http://drive5.com/uparse/, Edgar, R.C., Tiburon, CA, USA), and the single sequence and chimera were removed in the process of clustering. The sequence with the highest abundance in each OTU was annotated for species classification in the NCBI database (http://www.ncbi.nlm.nih.gov/) by RDP classifier (http://rdp.cme.msu.edu/). The comparison threshold was 70%. R software was used to conduct analysis and mapping of the community structure according to the taxonomic information corresponding to each OTU.

Based on the results of OTU clustering analysis, the coverage, Chao1, and Shannon’s diversity index [18] were calculated for each sample by using QIIME software (version 2.0, http://qiime.org/, Rob Knight Lab, Boulder, CO, USA). According to the total sequence number in the OTU abundance matrix, the rarefaction curve [19] was plotted using QIIME software, to compare the richness of the microbial community in the sediment samples.

To investigate the similarity of community structure among different samples, the community data structure was naturally decomposed by using QIIME software and R software through principal coordinates analysis (PCoA) [20] and the unweighted pair group method with arithmetic mean (UPGMA) [21]. The sample ordination was sequenced, to clarify the time and space differences in diversity and composition of *gcd* in sediment samples.

### 2.5. Statistical Analysis

Variance inflation factor (VIF) was used to screen the environmental factors to obtain those factors uncorrelated with each other [22]. Pearson correlation analysis was performed to determine whether significant correlations existed between *gcd* gene richness and the abovementioned environmental factors, and between *gcd* diversity and the abovementioned environmental factors. Analyses were performed using SPSS 20.0 (IBM, Armonk, NY, USA) for Windows. Significance levels were set at *p* = 0.05 and extreme significance levels at *p* = 0.01 in all statistical analyses. Canonical correlation analysis (CCA) was applied to evaluate the effect of environmental factors on the *gcd*-harboring bacterial community structure and was carried out with R language vegan package [23].

## 3. Results and Discussion

### 3.1. Physicochemical Properties of the Sediments and Overlying Water

The determination of physical and chemical factors of the sediments and overlying water in the spring and summer are shown in Table 2. As can be seen from Table 2, the pH of the overlying water showed little change among different seasons and different sampling locations. The temperature of the overlying water showed little difference between spring and summer for water stratification in Sancha Lake. However, the temperature was higher in L6, at the intake water area; L2, at the tail water area; and L4, near the area with intense human activity. DO was higher in the spring than in summer for all locations. In space, DO was higher in shallow water areas such as L6, L1, L2, and L4. DO in the dam was the lowest for deep water. The change trend of DTP was consistent with DO. TOC content in sediments showed little difference between spring and autumn. TN content in the spring was a little higher than in autumn, but the change of NH_3_–N content was the opposite. The contents of TP, IP, OP, HCl–P, and NaOH–P were all higher in the spring than in autumn. In space, all environmental factors except NH_3_–N were related with the intensity of human activity. The contents of TP, IP, OP, HCl–P, and NaOH–P were the highest at L1 and L3, in highly dense enclosure cultures; second highest at L5, in a dense enclose culture; and at L4, near an intense human activity area. The lowest was at L6 in the incoming water area.

### 3.2. OTU Classification and Determination and Alpha Diversity of the gcd Gene in Sediments

Total DNA was extracted from the microorganisms in sediment samples, and *gcd* gene sequences were acquired by PCR amplification. A total of 219,778 high-quality sequences were obtained by high-throughput sequencing, with an average length of 319 bp. The rarefaction curve of the sediment samples tended to be flat, indicating that the sequencing data volume was reasonable. Figure 3 shows the rarefaction curve diagram. Divided by 97% similarity, 610 OTUs were obtained. Table 3 shows that the coverage of each sample library was 99.58–99.94%, suggesting that the probability of *gcd* gene sequence detection in the sediment was very high. The sequencing results can represent different types of *gcd* genes and population types in the sediment samples of Sancha Lake.

The Chao1 index was used to estimate the total number of OTUs contained in the samples, reflecting the abundance of the *gcd*-harboring bacterial community. The larger Chao1, the higher abundance of the *gcd*-harboring bacterial community. Shannon’s index reflects the alpha diversity index of the bacterial community. The larger the Shannon value, the higher diversity of the bacterial community. There were spatial and temporal differences in the abundance and diversity of the gene *gcd* in the sediment of Sancha Lake (Table 3). The *gcd*-harboring bacterial community abundance (OTUs, Chao1) and diversity index (Shannon) were both higher in the spring than in autumn. The abundance of *gcd*-harboring communities was not exactly consistent with the sequence of diversity in the sampling sites in the spring and autumn. The highest abundance and diversity of *gcd* was at L6, in the incoming water area, and the lowest was at L3, in the center of the lake. In general, from the lake center to the lake dam and lake end, the biodiversity index and abundance of *gcd* showed an upward trend. The biodiversity and abundance of *gcd* at seriously polluted sampling points were even lower. The *gcd* biodiversity index in the spring was higher than that in the autumn.

RDP classifier was used to annotate the sequence with the highest abundance in each OTU, and the corresponding taxonomic information of each OTU was obtained by comparison with the NCBI database. In the sediment samples collected from Sancha Lake, a total of 46 genera of the *gcd* community with confirmed classification information were detected, which were subordinate to 29 families, 15 orders, nine classes, and six phyla. Compared with the NCBI database, there were still about 30,000 sequences that had no clear classification information, and about 20% of the *gcd*-harboring bacterial communities could not be identified. Therefore, the diversity of *gcd*-harboring bacterial communities in Sancha Lake sediments may actually be higher, containing many new species which require further exploration.

Bergkemper [9] et al. obtained six phyla, 17 orders, and 24 families by amplifying *gcd* from German forest soil samples. Rhizobiales, Rhodospirillales, and Burkholderiales (4%) were the dominant orders. Rodriguez found that Burkholderiales is an important inorganic phosphorus-soluble phosphate bacterial group [24]. In this study, six phyla, 15 orders, and 29 families were obtained. Rhizobiales, Oceanospirillales, and Burkholderiales (11.08%) were the dominant orders. At the family classification level, the functional bacteria of Burkholderiales, with the function of dissolving mineral phosphate, were more abundant and played an important role in the Sancha Lake eutrophication.

### 3.3. Changes in Composition of gcd-Harboring Bacterial Communities

#### 3.3.1. Comparison of *gcd*-Harboring Bacterial Community Composition Based on Classification Level

Six phyla of *gcd*-harboring bacterial communities were identified including *Proteobacteria*, *Acidobacteria*, *Gemmatimonadetes*, *Planctomycetes*, *Bacteroidetes*, and *Verrucomicrobia*. The relative abundance ratio greater than 1% was Proteobacteria (98.85–25.10%) and Acidobacteria (0–3.99%). Most reads came from Proteobacteria, which was the dominant group of *gcd* gene sources in sediments of the Sancha Lake. The reads of Proteobacteria were detected in each sample in two seasons and the number of ordinal reads in the spring was higher than that in autumn. Acidobacteria was the second dominant group, but was detected only at L1, L2, and L6 in the spring. *Gcd*-harboring bacterial community composition and the relative abundance in phylum level are shown in Figure 4. Please note that there were two unidentified taxa at the phylum level in the sediment samples. The abundance of one of them was 1.15–71.26%, and the number of reads was only second to Proteobacteria. It was detected in each sample in the spring and autumn and the number of reads in autumn was higher than that in spring. It was the dominant phylum but could not be identified. The other unidentified one was detected only at L6 in the spring, with an abundance of 0.35%. These two unidentified taxa indicated that there were many new species in the bacterial community of Sancha Lake sediments.

Twenty-one of 46 identified genera had more than 1% relative abundance. The distribution of relative abundance of more than 1% at the level of genera is shown in Figure 5. *Rhizobium* (1.63–77.99%), *Ensifer* (0.13–56.95%), *Shinella* (0.32–25.49%), and *Sinorhizobium* (0.16–11.88%) were detected in each sample from all locations both in spring and autumn. Their relative abundance was higher in the spring than in autumn and higher at L1, L2, and L4, where eutrophication was heavy. It is supposed that these bacteria have great effect on lake eutrophication. The relative abundance of the four genera was significantly different at all sites. *Agrobacterium* (0–72.92%) was detected at all sites except L3, which was in a deep-water area both in the spring and autumn, and its relative abundance was high. *Enterobacter* (0–1.12%) was detected at all sites only in the spring, and *Paracoccus* (0–1.15%) was detected at all sites only in autumn.

The distribution of *gcd*-harboring bacterial communities was different in different seasons and different sediment types. The diversity of community composition from phyla to genera increased gradually, but the difference of population from phyla to genera was not significant. The dominant bacterial communities with higher relative abundance could be detected in different seasons and sediment types. The abundance of *gcd*-harboring bacteria was higher at heavy eutrophication sites. Some bacterial communities were unique in time or space.

#### 3.3.2. Analysis of the Composition of *gcd*-Harboring Bacterial Communities Based on OTU Level

Clustering and sequencing were performed according to the abundance distribution of OTU or the similarity between samples. R software was used for cluster analysis and drawing a heatmap diagram of the top 30 OTUs. By clustering, high- and low-abundance OTUs could be distinguished. The similarity and difference of community composition between samples is reflected by the color gradient. Red indicates high out abundance, while blue indicates low OTU abundance, which directly reflects the similarity and difference in the composition of the bacterial community with high abundance in the sediment of Sancha Lake.

Heatmap analysis of *gcd*-harboring bacterial community composition is shown in Figure 6. It can be seen from the gradient change of color that the high-abundance *gcd*-harboring bacterial communities from L1, L2, L3, and L5 in spring and autumn are in one group, indicating that the seasonal change had little influence on the high-abundance *gcd*-harboring bacterial communities. However, *gcd*-harboring bacterial communities from L4 and L6 in the spring and autumn did not gather in the same group, and the composition of the high-abundance *gcd*-harboring bacterial community had obvious seasonal variation. This suggests that seasonal variation has a great impact on the distribution of the high-abundance *gcd*-harboring bacterial community of L4 and L6. In space, it can be seen that the highly enriched *gcd*-harboring bacterial communities in the sediments of the six sampling sites did not cluster with each other. The spatial differences of the bacterial community showed significant changes, indicating that the sampling site type had a great influence on the high abundance of *gcd*-harboring bacterial communities, resulting in the heterogeneous degree of Sancha Lake eutrophication.

#### 3.3.3. Comparative Analysis of *gcd*-Harboring Bacterial Community Composition Based on Beta Diversity

The distance matrices of weighted and unweighted UniFrac were calculated by QIIME, and UPGMA and PCoA were analyzed, using R language, as a picture tree to further evaluate the similarity and difference of the composition of *gcd*-harboring bacterial communities in different seasons and sample sites of sediment in Sancha Lake. As can be seen from Figure 7 and Figure 8, for seasonal change, the *gcd*-harboring bacterial communities from L1 to L3 and L5 in the spring and autumn have their branches or gather in the same quadrant. Regardless of the season, they all came together, indicating that seasonal variation had little effect on the difference of the composition of *gcd*-harboring bacterial communities in these four sampling sites.

The *gcd*-harboring bacterial communities from L4 and L6 changed greatly in the spring and autumn, forming different branches or not being in the same quadrant, respectively. Seasonal variation had a great influence on the difference in the composition of the *gcd*-harboring bacterial communities in L4 and L6. This may be attributed to the varying intensity of human activity. L6 was in the incoming water area and L4 was near an area of intense human activity. In the process of spatial change of each sampling site, three groups formed. Among them, L2 formed a group, L1 and L3 constituted a group, and L5 made up another group. As the seasons changed, L4 and L6 were located in them, which was consistent with the characterization of the relatively high-abundance heatmap.

Based on the analysis of the composition of *gcd*-harboring bacterial communities from the classification level, OTU level, and beta diversity, it was found that the diversity and composition of *gcd*-harboring bacterial communities in the sediments of Sancha Lake presented obvious temporal and spatial changes. The diversity and abundance of the *gcd*-harboring bacterial communities in spring were higher than that in autumn, but the difference of the same sample site was different with seasonal changes. In space, because L6 was in the main water entry area of the lake, the water was highly fluid. By contrast, L1 and L3 were in the relatively concentrated area of fenced breeding. L2 was in the tail water area of the reservoir, L5 was in the dense area of cage breeding, and L4 was in the adjacent area of intensive human activities. The variations of time and place of feeding in the Sancha Lake resulted in different pollution zones at the bottom of the lake. Furthermore, the difference in the degree of human activity interference and water flow affected the difference in the composition of *gcd*-harboring bacterial communities.

### 3.4. Correlation between gcd-Harboring Bacterial Communities and Environmental Conditions in the Sediment of Sancha Lake

#### 3.4.1. Linking *gcd*-Harboring Bacterial Abundance, Diversity, and Community Structure with Environmental Factors

The environmental factors with VIF greater than 10 were filtered out by the VIF method, and multiple screenings were conducted until all the corresponding VIF values of the selected environmental factors were less than 10. Pearson correlation analysis was performed for environmental factors and showed small interactions after screening; the correlation analysis between the two factors is shown in Table 4. The results showed that the Simpson index of *gcd* genes was significantly negatively correlated with the concentration of TOC, TP, and HCl–P (*p* < 0.05) and was extremely significantly positively correlated with the concentration of DTP (*p* < 0.01). The Chao1 index was significantly negatively correlated with the concentration of TOC, TN, TP, and HCl–P (*p* < 0.05) and was positively correlated with DTP concentration (*p* < 0.05). Changes in effective reads were significantly positively correlated with DTP concentration (*p* < 0.05). The number of observed OTUs had a significant negative correlation with the change of T (*p* < 0.05) and showed a significant positive correlation with DTP concentration (*p* < 0.01). In general, the diversity and abundance of *gcd* genes were not strongly correlated with pH and T indicators in water and were closely related to physical and chemical factors such as the concentrations of DO, TOC, TN, TP, and DTP.

#### 3.4.2. Correlation between Spatial Distribution Characteristics of *gcd*-Harboring Bacterial Communities and Environmental Conditions

Based on the OTU level, the correlation between the spatial distribution of *gcd* genes in sediments and environmental conditions was analyzed by RDA/CCA using rda or cca in the vegan package of R language. In the discriminant component analysis (DCA) by species-sample database, the length of the first axis of lengths of gradient was greater than 4.0. Thus, CCA (Jari, k., University of Oulu at Oulu, Finland) was selected to analyze the correlation between the *gcd* group and environmental factors, to clarify the correlation between the spatial and temporal distribution of the *gcd*-harboring bacterial communities in the sediments of Sancha Lake and environmental factors. Environmental factors were screened by the VIF method, and CCA analysis was conducted at the OTU level. The analysis results are shown in Figure 9.

The resolution of the *X*-axis and *Y*-axis were 15.83% and 13.72%, respectively. The physical and chemical factors of the sediments are represented by a line with an arrow, and the length of the ray represents the degree of influence of the factor on the community OTU. The quadrant in which the arrow is located represents the positive and negative correlations between the factor and the sorting axis, and the angle between the arrow line and the sorting axis represents the correlation between the factor and the sorting axis. When the angle between physicochemical factors of sediment is acute, it indicates that the two environmental factors are positively correlated, while a pure angle means they are negatively correlated.

Among all samples in two seasons, the distribution of samples was relatively dispersed, and the influence of environmental factors did not make the spatial and temporal distribution of sample sites show obvious regularity. However, in terms of the overall relationship between OTU and environmental factors, both TOC and HCl–P had long on-lines on the one and two sequencing axes, and the results show that the influence of sediment TOC and HCl–P on the OTU of the whole *gcd*-harboring bacterial community was very significant, and they had an extremely significantly negative correlation (*p* < 0.01). The change of the overlying water DTP content was positively correlated with the OTU of the whole *gcd*-harboring community.

Zeng Qingwei et al. found that the phosphorus-soluble activity of the inorganic phosphorus-soluble phosphor bacteria and the content of soluble phosphorus (DTP) in the environment influence each other [7] and thus speculated that the change of DTP content in the overlying water of Sancha Lake was partly caused by the secretion of organic acids by the *gcd*-harboring bacterial community to dissolve inorganic phosphorus. The release of DTP in the overlying water indicates that the *gcd*-harboring bacterial community in sediments has a certain effect on the eutrophication of water. HCl–P and NaOH–P were negatively correlated with OTUs of *gcd*-harboring bacterial communities. Compared with NaOH–P and HCl–P, HCl–P was more significantly correlated, indicating that the *gcd*-harboring bacterial community may give priority to HCl–P. Among the three physical factors of DO, T, and pH, DO had a significantly positive correlation with OTU spatial and temporal distribution of the *gcd*-harboring bacterial community, while T and pH had no significant correlation.

## 4. Conclusions

A total of 219,778 high-quality sequences were obtained from sediment samples of Sancha Lake by DNA extraction, gene *gcd* sequence amplification with PCR, and Illumina MiSeq platform sequencing. The *gcd*-harboring bacterial communities with confirmed classification information were composed of six phyla, nine classes, 15 orders, 29 families, and 46 genera. The *gcd* genes had a higher diversity. About 20% of the *gcd*-harboring bacterial communities could not be identified. Therefore, the genetic *gcd* diversity in the sediments of Sancha Lake may actually be higher and many new species may exist. From the center of the lake to the dam, and then to the end of the lake, the biodiversity index and abundance showed an upward trend. The *gcd* diversity index and abundance in the seriously polluted sampling sites were even lower. The bacterial diversity index of *gcd* in the spring was higher than that in autumn. The abundance of these dominant *gcd*-harboring bacteria, such as *Rhizobium*, *Ensifer*, *Shinella*, and *Sinorhizobium*, were higher in the spring than in autumn, suggesting that they have an important effect on the eutrophication of Sancha Lake. The major environmental factors affecting *gcd* gene diversity and *gcd*-harboring bacterial community composition were determined to be DO, TOC, TN, TP, HCl–P, and DTP through Pearson correlation analysis and CCA analysis. The results indicated that GDH encoded by *gcd* genes catalyzed the oxidation of glucose into gluconic acid, resulting in insoluble phosphate transformation into soluble phosphate in the sediments. DTP concentration in overlying water increased and accelerated the eutrophication of Sancha Lake. Thus, *gcd* genes encoding GDH play an important role in lake eutrophication.

## Figures and Tables

**Figure 1 ijerph-16-00001-f001:**
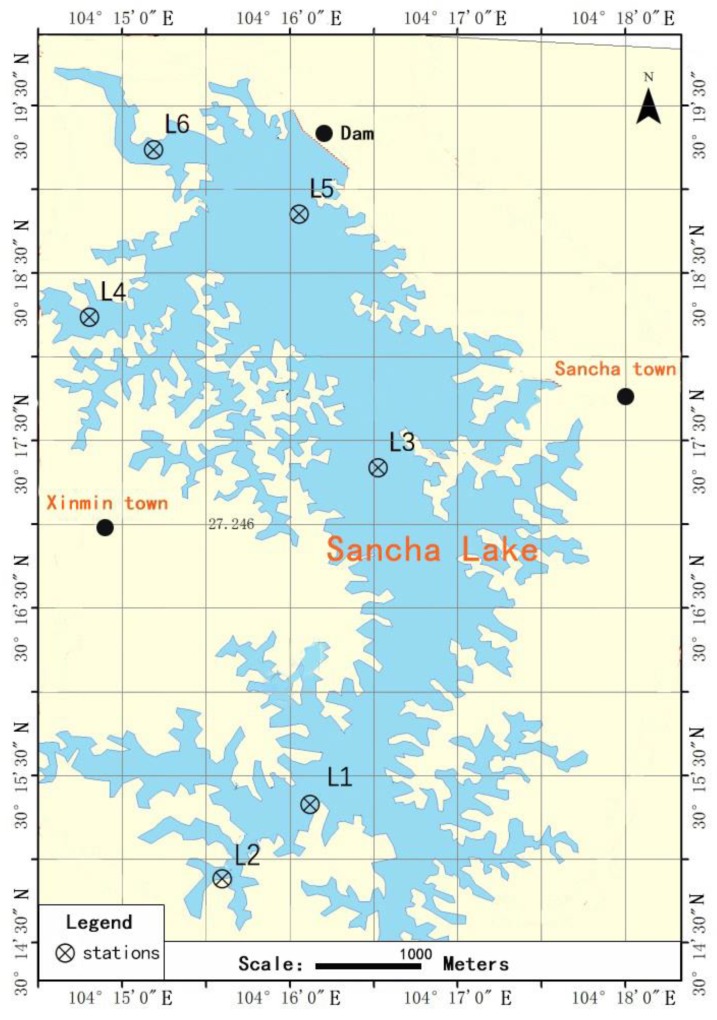
Sampling sites at Sancha Lake.

**Figure 2 ijerph-16-00001-f002:**
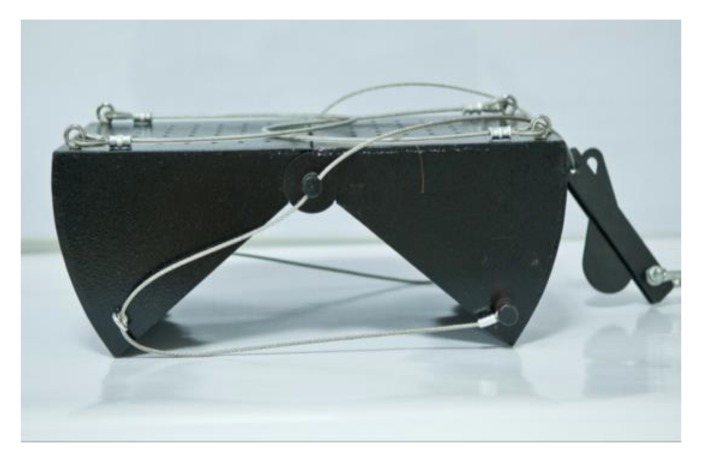
Claw-like Peterson dredge.

**Figure 3 ijerph-16-00001-f003:**
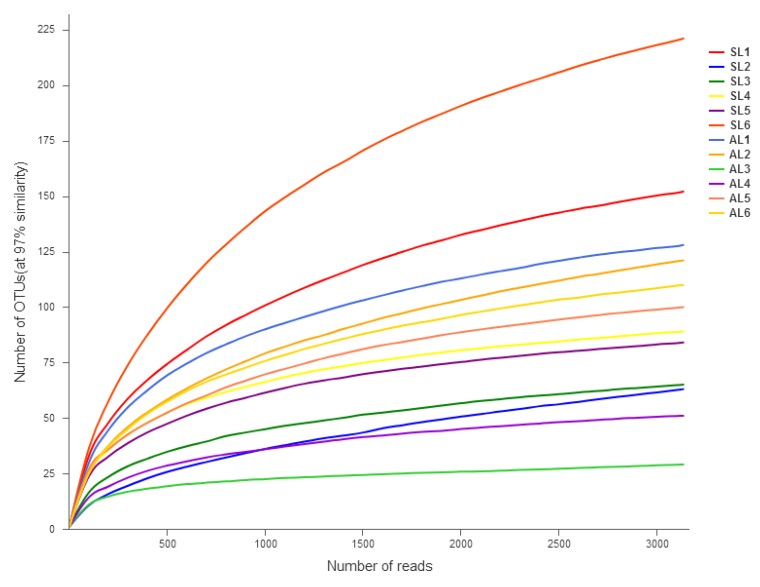
Rarefaction curves of samples L1–L6 in the spring and autumn.

**Figure 4 ijerph-16-00001-f004:**
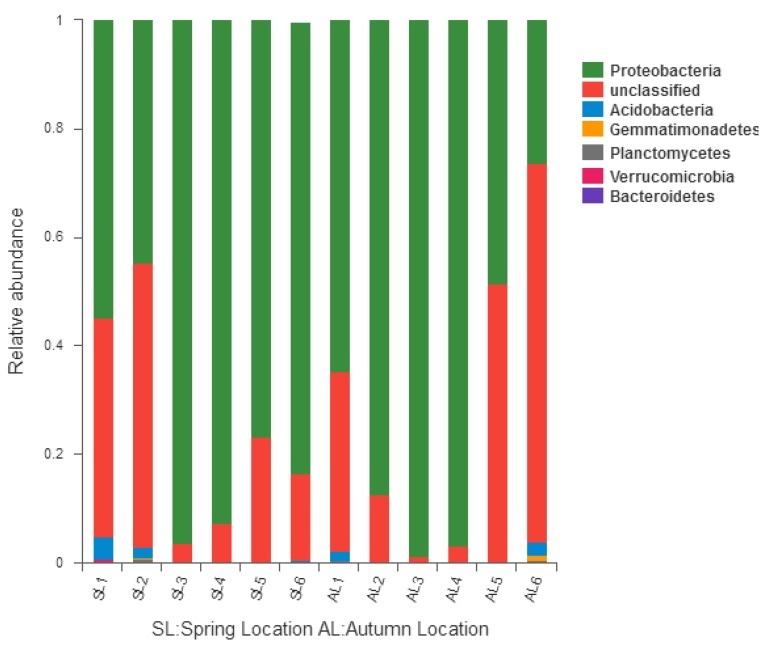
Relative abundance and composition of *gcd*-harboring bacterial phyla detected in the sediments of Sancha Lake.

**Figure 5 ijerph-16-00001-f005:**
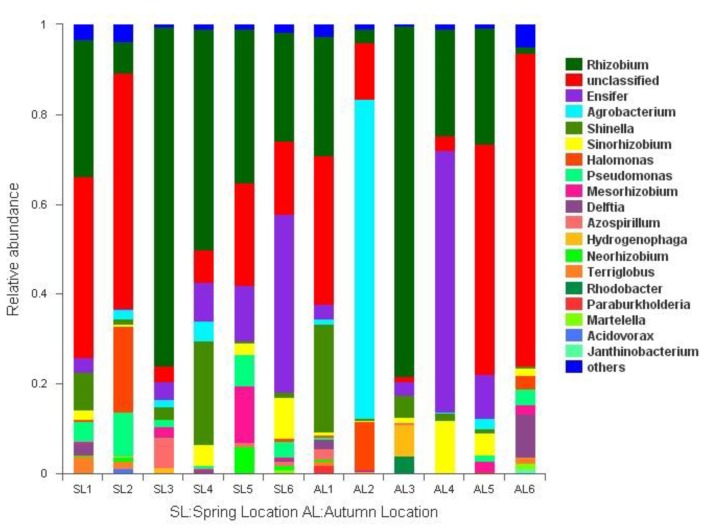
Relative abundance and composition of *gcd*-harboring bacterial genera detected in the sediments of Sancha Lake.

**Figure 6 ijerph-16-00001-f006:**
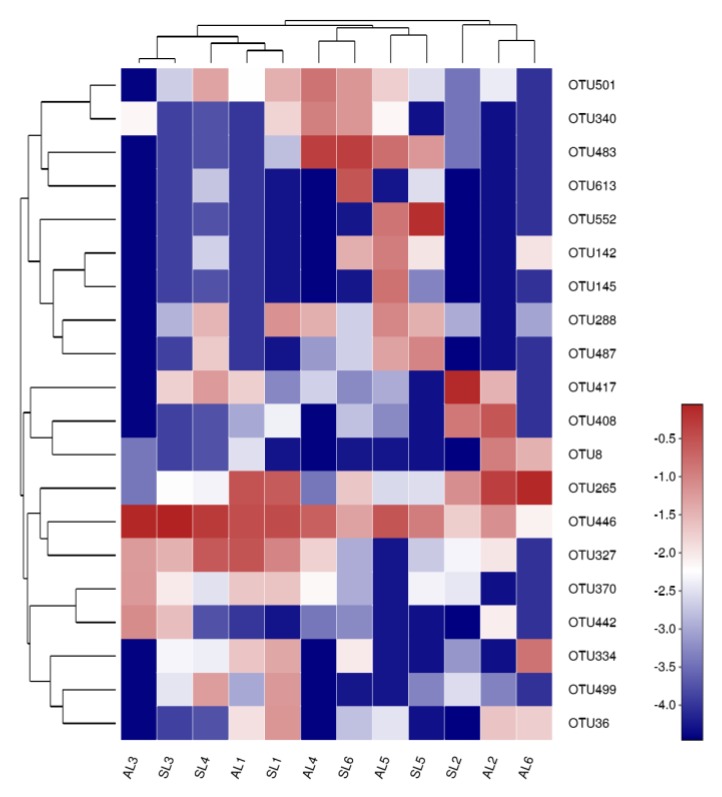
The heatmap diagram of *gcd*-harboring bacterial communities in the sediments of Sancha Lake.

**Figure 7 ijerph-16-00001-f007:**
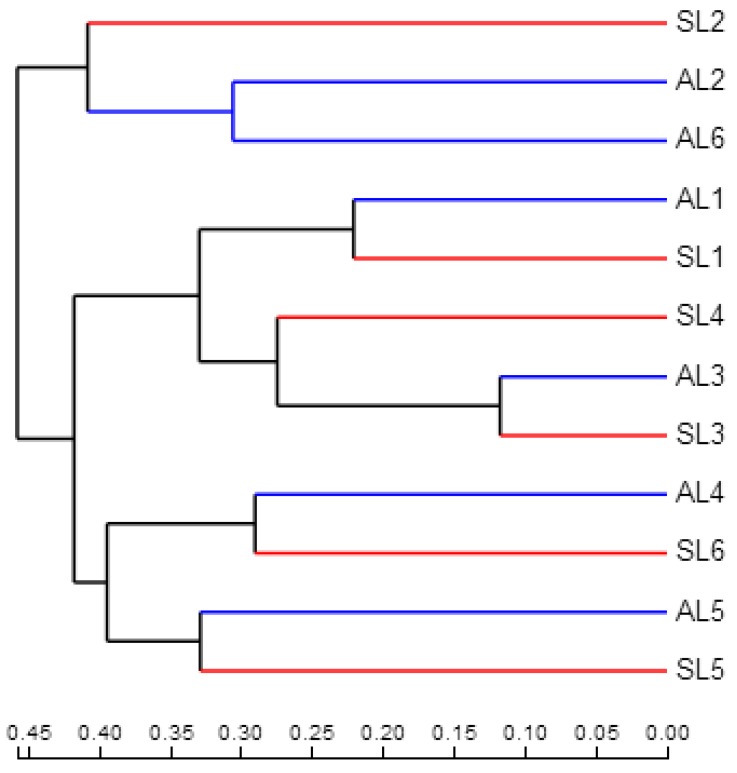
The UPGMA analysis of *gcd*-harboring bacterial communities in the sediments of Sancha Lake.

**Figure 8 ijerph-16-00001-f008:**
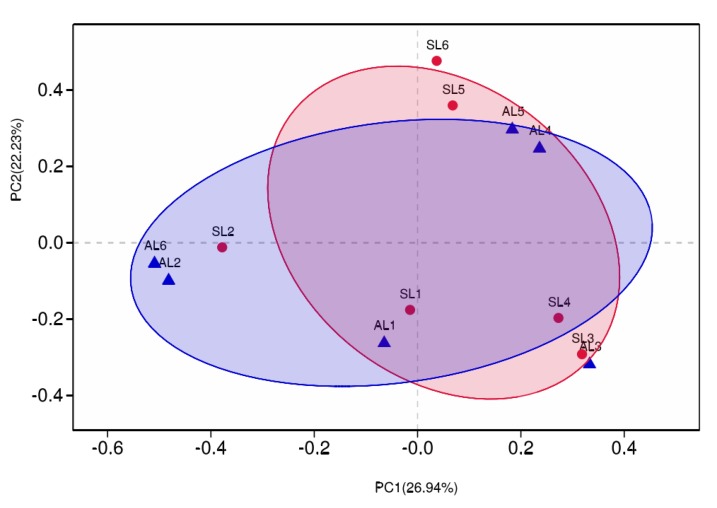
The PCoA analysis of *gcd*-harboring bacterial communities in the sediments of Sancha Lake.

**Figure 9 ijerph-16-00001-f009:**
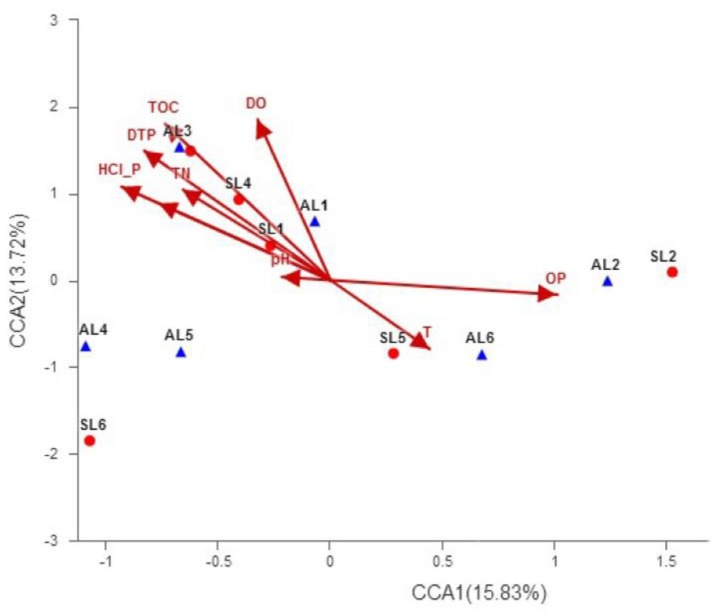
The CCA analysis of *gcd*-harboring bacterial communities and physicochemical factors of the sediments in Sancha Lake.

**Table 1 ijerph-16-00001-t001:** Description of sampling sites at Sancha Lake.

Sample Site	Geographical Coordinates	Depth (m)	Hydrophyte	Description
L1	30°14′52″ N	13	Large quantity	Concentrated area of fenced breeding
	104°16′15″ E			
L2	30°14′28″ N	4	Large quantity	Tail water area of the reservoir
	104°15′32″ E			
L3	30°17′25″ N	26	Small quantity	Relatively concentrated area of fenced breeding
	104°16′31″ E			
L4	30°18′15″ N	17	Large quantity	Area with intense human activity
	104°14′31″ E			
L5	30°18′18″ N	30	Small quantity	Dense area of cage breeding
	104°16′2″ E			
L6	30°19′15″ N	19	Moderate quantity	Main water entry area of the lake
	104°15′14″ E			

**Table 2 ijerph-16-00001-t002:** Physicochemical properties of the sediments and overlying water in spring and autumn.

Season	Location	pH	DO (mg·L^−1^)	T (°C)	DTP (mg·L^−1^)	TOC (mg·g^−1^)	TN (mg·g^−1^)	NH_3_-N (mg·g^−1^)	TP (mg·g^−1^)	OP	IP (mg·g^−1^)	HCl–P (mg·g^−1^)	NaOH–P (mg·g^−1^)
Spring	L1	7.38 ± 0.11	6.4 ± 1.0	13.0 ± 0.3	0.088 ± 0.025	48.1 ± 5.0	6.46 ± 0.46	0.387 ± 0.030	1.036 ± 0.100	0.242 ± 0.042	0.790 ± 0.090	0.590 ± 0.01	0.148 ± 0.008
	L2	7.52 ± 0.13	6.4 ± 0.9	12.9 ± 0.1	0.092 ± 0.040	36.9 ± 4.0	4.30 ± 0.30	0.057 ± 0.007	0.715 ± 0.015	0.229 ± 0.090	0.617 ± 0.017	0.550 ± 0.010	0.047 ± 0.010
	L3	7.52 ± 0.10	5.8 ± 0.2	12.6 ± 0.3	0.035 ± 0.005	76.6 ± 8.0	10.15 ± 1.00	0.017 ± 0.007	3.069 ± 0.092	0.562 ± 0.062	2.687 ± 0.087	2.248 ± 0.100	0.367 ± 0.060
	L4	7.46 ± 0.10	6.7 ± 1.1	13.2 ± 0.2	0.069 ± 0.002	55.0 ± 6.0	4.87 ± 0.87	0.021 ± 0.000	1.120 ± 0.120	0.189 ± 0.046	0.824 ± 0.004	0.786 ± 0.050	0.099 ± 0.009
	L5	7.45 ± 0.09	5.1 ± 0.5	12.6 ± 0.2	0.033 ± 0.003	55.6 ± 5.0	6.57 ± 0.50	0.067 ± 0.007	1.376 ± 0.109	0.311 ± 0.011	1.162 ± 0.10	0.556 ± 0.010	0.340 ± 0.020
	L6	7.67 ± 0.21	9.0 ± 1.0	12.7 ± 0.3	0.065 ± 0.005	25.4 ± 3.0	1.66 ± 0.06	0.096 ± 0.006	0.696 ± 0.100	0.099 ± 0.009	0.481 ± 0.090	0.406 ± 0.006	0.077 ± 0.007
Autumn	L1	7.54 ± 0.10	5.6 ± 0.1	15.4 ± 0.4	0.057 ± 0.007	47.5 ± 2.0	4.49 ± 0.49	0.244 ± 0.034	2.874 ± 0.074	0.317 ± 0.017	2.603 ± 0.060	1.088 ± 0.088	0.172 ± 0.002
	L2	6.89 ± 0.19	5.5 ± 0.1	17.2 ± 1.0	0.051 ± 0.002	34.8 ± 1.8	3.18 ± 0.28	0.035 ± 0.005	1.153 ± 0.053	0.444 ± 0.021	0.782 ± 0.030	0.450 ± 0.050	0.395 ± 0.005
	L3	7.11 ± 0.10	4.1 ± 0.1	14.6 ± 0.6	0.012 ± 0.001	83.2 ± 3.2	10.77 ± 1.74	0.539 ± 0.039	5.005 ± 0.202	0.609 ± 0.041	3.882 ± 0.080	3.517 ± 0.017	1.007 ± 0.007
	L4	7.16 ± 0.06	5.5 ± 0.2	16.3 ± 0.9	0.055 ± 0.005	54.2 ± 2.4	3.99 ± 0.41	0.141 ± 0.004	2.661 ± 0.060	0.295 ± 0.010	1.897 ± 0.070	1.949 ± 0.049	0.222 ± 0.022
	L5	7.02 ± 0.02	4.5 ± 0.1	14.1 ± 0.1	0.015 ± 0.005	47.8 ± 2.0	5.25 ± 0.79	0.071 ± 0.001	1.454 ± 0.054	0.369 ± 0.009	1.051 ± 0.051	0.554 ± 0.054	0.611 ± 0.011
	L6	7.18 ± 0.18	5.9 ± 0.3	16.5 ± 0.5	0.025 ± 0.005	32.9 ± 1.7	2.28 ± 0.18	0.265 ± 0.006	0.629 ± 0.029	0.167 ± 0.007	0.444 ± 0.024	0.343 ± 0.033	0.104 ± 0.004

Note: pH, T, DO, and DTP were measured in overlying water of sediments; TOC, TN, NH_3_–N, TP, IP, OP, HCl–P, and NaOH–P were measured in sediments. Data are mean ± standard deviation.

**Table 3 ijerph-16-00001-t003:** The *gcd*-harboring bacteria richness and diversity in sediments in Sancha Lake.

Season	Sampling Site	Reads	Chao1	Shannon	Coverage	OTUs	No. of Phyla	No. of Classes	No. of Orders	No. of Families	No. of Genera
spring	L1	32,585 ± 150	234.55 ± 33.05	3.365 ± 0.040	0.9993 ± 0.0004	226 ± 6	6	9	13	19	22
	L2	23,251 ± 120	156.18 ± 21.04	2.777 ± 0.067	0.9985 ± 0.0012	150 ± 4	5	9	12	15	23
	L3	12,834 ± 60	98.60 ± 19.04	1.539 ± 0.030	0.9986 ± 0.0007	85 ± 2	2	4	7	11	16
	L4	41,828 ± 200	185.27 ± 29.04	2.622 ± 0.064	0.9993 ± 0.0005	158 ± 4	1	3	5	10	17
	L5	12,151 ± 60	140.69 ± 18.09	3.167 ± 0.071	0.9983 ± 0.0002	130 ± 3	3	5	7	10	13
	L6	13,357 ± 65	322.14 ± 36.64	3.397 ± 0.075	0.9958 ± 0.0017	291 ± 7	3	6	9	17	32
autumn	L1	95,695 ± 50	163.05 ± 23.04	2.943 ± 0.040	0.9978 ± 0.0014	154 ± 4	5	8	12	16	22
	L2	10,880 ± 54	143.96 ± 20.05	1.259 ± 0.014	0.9983 ± 0.0004	121 ± 3	2	4	5	5	15
	L3	3199 ± 10	43.00 ± 1.00	1.307 ± 0.016	0.9974 ± 0.0000	29 ± 0	1	3	5	6	10
	L4	10,852 ± 54	80.20 ± 3.04	1.939 ± 0.029	0.9984 ± 0.0006	62 ± 2	1	3	5	7	10
	L5	30,857 ± 150	139.38 ± 17.04	2.432 ± 0.039	0.9994 ± 0.0001	118 ± 3	2	3	5	8	12
	L6	18,415 ± 92	157.53 ± 22.04	2.273 ± 0.027	0.9992 ± 0.0005	152 ± 3	5	8	13	21	23

Note: Data are mean ± standard deviation.

**Table 4 ijerph-16-00001-t004:** Coefficients of correlation between diversities and abundances of *gcd* genes in the sediments and physicochemical properties.

Environmental Factor	PH	DO	T	TN	TOC	TP	HCl-P	NaOH-P	DTP
OTUs	0.422 ^a^/0.172 ^b^	0.346/0.271	0.239/0.454	–0.577/0.05 *	–0.631/0.028 *	–0.698/0.012 *	–0.687/0.014 *	–0.624/0.03 *	0.915/0.000 **
Chao1	0.043/0.894	0.391/0.211	–0.063/0.846	–0.604/0.037 *	–0.6/0.039 *	–0.619/0.032 *	–0.653/0.021 *	–0.32/0.311	0.643/0.024 *
Shannon	479/0.115	0.251/0.432	0.278/0.382	–0.574/0.051	–0.619/0.032 *	–702/0.011 *	–0.672/0.017 *	–0.635/0.26	0.921/0.000 **
No. of filtered reads	0.371/0.236	0.907/0.002 **	0.12/0.711	–0.093/0.773	–0.191/0.553	–0.464/0.129	–0.457/0.135	–514/0.088	0.602/0.039 *

^a^*R*^2^; ^b^*p*-value; * *p* < 0.05, ** *p* < 0.01.

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
