# Peer review of "Gcd Gene Diversity of Quinoprotein Glucose Dehydrogenase in the Sediment of Sancha Lake and Its Response to the Environment"

_ijerph, 2018, doi:10.3390/ijerph16010001_

Round 1

Reviewer 1 Report

In the presented work, the authors studied diversity of GDH gene gcd to identify the microorganism group that significantly influence on eutrophication of Sancha Lake water.  The manuscript has a potentially interesting premise which could add to the knowledge base of gcd gene diversity of GDH and its role in eutrophication of lake water. Overall, the work is of potential interest for publication; however the following points need to be addressed first.

1.     The English grammar, formatting needs to be redone throughout the manuscript. The basic grammar and writing format is also not followed throughout.  For example line 39: causing eutrophication of water.Among “need space after water. Among” all these formatting issues are throughout the manuscript.  The authors at least need to know the journal writing format.

2.     In introduction section the authors has not included any sufficient  literature  review on researches conducted on diversity of gcd genes, their outcomes and why this research was needed to conduct and how this research helps on gaping the data gaps. In other word “the author should be able to clarify what’s the hypothesis of their research and objectives that are different from any previous studies conducted”.  Also, the introduction section fails to provide insights on previous studied on diversity of gcd genes, need for this study and added values from the result of this study.  I recommend authors to clarify all these above mentioned comments. Line 59: Describe the acronyms DO, TOC TP, DTP for authors.

3.     Site Description and  Sample Collection Section:

a) I recommend presenting a table providing the site characteristics.

b) Line 70-71: Please explain “the inflow of CODcr and BOD5 shows downward trend year by year…….” what does this explain to readers? What is the hypothesis for selection of six sampling point? Please explain.

c) Line 75: it should be Figure. 1 or Fig. 1 not FIG.  Also can you present the figure of claw like Peterson dredge? Was any specific sampling protocol or standard methods followed for taking samples and sample preparation please include the standard methods

4.  Line 89: Describe the acronyms IP, OP, NAOH-P, HCI-P.

5. Line 100: Explain “Three parallel samples were taken” does it mean duplicates and triplicate samples? Please explain in brief the sampling protocol followed as mentioned previously

6. Line 102: “Because the sediment was of high water content “ where is this result shown its confusing explain.

7. Line 108-109: to many spaces formatting issues

8. Line 147: For the measured environmental factors, …………………………….with little interaction” Explain what does this mean

9. Line 159: “As can be seen from Fig. 1, pH of the overlying…….” I can’t see any pH values in Figure 1. Do you mean Table 1?

10. Table 1. Fix the formatting issues and show the standard deviation values for all physicochemical properties.

11. Table 2: Same as above.

12: Figure 4: this figure is too busy to provide the information can you present in any other format if possible or a table instead of figure.

13: Conclusion: The authors have just mentioned the results in conclusion section. I don’t find any conclusion of the research conducted presented here and its impact on the scientific research field. I recommend authors to include the finding of the research and its conclusion here.

Author Response

Dear Editors and Reviewers:

Revised portion are marked using the "Highlight" function in the paper. The main corrections in the paper and the responds to the reviewer’s comments are as flowing:

Responds to the reviewer’s comments:

Comments and Suggestions for Authors

Comments and Suggestions for Authors

In the presented work, the authors studied diversity of GDH gene gcd to identify the microorganism group that significantly influence on eutrophication of Sancha Lake water.  The manuscript has a potentially interesting premise which could add to the knowledge base of gcd gene diversity of GDH and its role in eutrophication of lake water. Overall, the work is of potential interest for publication; however the following points need to be addressed first.

1.     The English grammar, formatting needs to be redone throughout the manuscript. The basic grammar and writing format is also not followed throughout.  For example line 39: causing eutrophication of water.Among “need space after water. Among” all these formatting issues are throughout the manuscript.  The authors at least need to know the journal writing format.

 Reply: The English grammar and writing format has been redone and much improved to make it easier to understand.

2.     In introduction section the authors has not included any sufficient  literature  review on researches conducted on diversity of gcd genes, their outcomes and why this research was needed to conduct and how this research helps on gaping the data gaps. In other word “the author should be able to clarify what’s the hypothesis of their research and objectives that are different from any previous studies conducted”.  Also, the introduction section fails to provide insights on previous studied on diversity of gcd genes, need for this study and added values from the result of this study.  I recommend authors to clarify all these above mentioned comments. Line 59: Describe the acronyms DO, TOC TP, DTP for authors.

Reply:1)The hypothesis and objectives of our research are clarified as follows:

Our aims were to define what gcd gentic backgrounds were generated in lake sediments, and what factors were related to gcd genetic distributionso as to reveal the relationships of lake eutrophication and gcd-harboring bacterial communities.

We hypothesized that gcd-harboring bacteria were diverse and varied for different seasons and different sampling site types. Gcd genes encoding GDH were closely related to lake eutrophication.

2) Insights of previous studies on the diversity of gcd genes were added as follows:

 All these studies didn’t cover gcd gene diversity. No researches have been conducted on the diversity of gcd genes and its relationship with eutrophication in lake sediments.

3) Values of our study were added as follows:

This paper studied the diversity, the spatial and temporal distribution of gene gcd in Sancha Lake sediments and its response to environmental factors such as DO, TOC, TP and DTP by DNA extraction of sediments, PCR amplification and high-throughput sequencing in order to explore gcd gene diversity and its response to environment, which is useful to control eutrophication.

4) The acronyms DO, TOC TP, DTP were describled as follows:

dissolved oxygen (DO), total organic carbon (TOC), total phosphorus (TP), dissolved total phosphorus (DTP).

3.     Site Description and  Sample Collection Section:

a) I recommend presenting a table providing the site characteristics.

Reply:The characteristics information of all sampling sites is presented in Table 1

Table 1. Description of sampling sites in the Sancha Lake.

Sample

Site

Geographical

coordinates

Depth

(m)

hydrophyte

Description

L1

30˚14′52″N

104˚16′15″E

13

Large quantity

Concentrated area of fenced breeding

L2

30˚14′28″N

104˚15′32″E

4

Large quantity

Tail water area of the reservoir

L3

30˚17′25″N

104˚16′31″E

26

Small quantity

Relatively concentrated area of fenced breeding

L4

30˚18′15″N

104˚14′31″E

17

Large quantity

The area with intense of human activity

L5

30˚18′18″N

104˚16′2″E

30

Small quantity

The dense area of cage breeding

L6

30˚19′15″N

104˚15′14″E

19

Moderate quantity

Main water entry area of the lake

b) Line 70-71: Please explain “the inflow of CODcr and BOD5 shows downward trend year by year…….” what does this explain to readers? What is the hypothesis for selection of six sampling point? Please explain.

Reply: The part “the inflow of CODcr and BOD5 shows downward trend year by year…….” Was explained as follows:

According to the monitoring results over years, the inflow of CODCr and BOD5intothe Sancha Lakeshows a downward trend year by yearfor the control of external pollutants. Butthe TN and TP of water body has been on an upward trend. The corresponding chlorophyll increases on a yearly basis while the transparency decreases year by year, and the water has been eutrophicated [9]. The Sancha Lake eutrophication might be attributed to insoluble phorsphate transformed to soluble phorsphate by gcd-harboring bacterial communites.

c) Line 75: it should be Figure. 1 or Fig. 1 not FIG.  Also can you present the figure of claw-like Peterson dredge? Was any specific sampling protocol or standard methods followed for taking samples and sample preparation please include the standard methods

1) FIG has been corrected.

2) The figure of claw-like Peterson dredge is presented:

Fig. 2 claw-like Peterson dredge figure

3)Sampling protocol and sample preparation were referring to the study by Liu et al, 2014:

At each samping site, there are three sampling points. In April (spring) and November (autumn), 2017, claw-like peterson dredge was used to capture surface sediments at each sampling point (Fig. 1). The surface (0-125px) sediment was collected by plexiglass column and put into a clean sealed polythene bag[10]. Three parallel samples were collected at each sample point and mixed as the representative sample of the sample point. Some sediments were stored in 4℃ for physical and chemical analysis (within24 h), and some sediment samples were stored in -80℃ for DNA extraction. At the same time, the air-tight water sampler was used to collect the overlying water above the sediment layer at each sampling point for the analysis of water environment index[10].

4.  Line 89: Describe the acronyms IP, OP, NAOH-P, HCI-P.

Reply: The acronyms IP, OP, NAOH-P, HCI-P were inorganic phosphorus (IP), organic phosphorus (OP), phosphonium hydroxide (NaOH-P), phosphorus hydrochloride (HCl-P).

5. Line 100: Explain “Three parallel samples were taken” does it mean duplicates and triplicate samples? Please explain in brief the sampling protocol followed as mentioned previously.

Reply: The sentence “Three parallel samples were taken” here should be deleted.

6. Line 102: “Because the sediment was of high water   “ where is this result shown its confusing explain.

Reply: The sentence “Because the sediment was of high water…”is corrected as follows:

The total DNA was extracted after centrifugation to remove excess water from the sediment.

7. Line 108-109: to many spaces formatting issues

Reply: Format issues have been fixed.

8. Line 147: For the measured environmental factors, with little interaction” Explain what does this mean

The sentence“For the measured environmental factors, with little interaction” has been corrected as follows:

“Variance Inflation FactorVIFwas used to screen the environmental factors to get those factors uncorrelated with each other [21].”

9. Line 159: “As can be seen from Fig. 1, pH of the overlying…….” I can’t see any pH values in Figure 1. Do you mean Table 1?

Reply: “Fig. 1” has been corrected to “Table 1”.

10. Table 1. Fix the formatting issues and show the standard deviation values for all physicochemical properties.

 Reply: Formatting issues of table 2 have been fixed and standard deviation values have been shown in table 2.

Table 2. Physicochemical properties of the sediments and overlying water in spring and autumn.

Season

Location

pH

DO mg·L-1

T

(℃)

DTP mg·L-1

TOC mg·g-1

TN mg·g-1

NH3-N

mg·g-1

TP

mg·g-1

OP

IP

mg·g-1

HCl-P

mg·g-1

NaOH-P

mg·g-1

spring

L1

7.38±0.11

6.4±1.0

13.0±0.3

0.088±0.025

48.1±5.0

6.46±0.46

0.387±0.030

1.036±0.100

0.242±0.042

0.790±0.090

0.590±0.01

0.148±0.008

L2

7.52±0.13

6.4±0.9

12.9±0.1

0.092±0.040

36.9±4.0

4.30±0.30

0.057±0.007

0.715±0.015

0.229±0.090

0.617±0.017

0.550±0.010

0.047±0.010

L3

7.52±0.10

5.8±0.2

12.6±0.3

0.035±0.005

76.6±8.0

10.15±1.00

0.017±0.007

3.069±0.092

0.562±0.062

2.687±0.087

2.248±0.100

0.367±0.060

L4

7.46±0.10

6.7±1.1

13.2±0.2

0.069±0.002

55.0±6.0

4.87±0.87

0.021±0.000

1.120±0.120

0.189±0.046

0.824±0.004

0.786±0.050

0.099±0.009

L5

7.45±0.09

5.1±0.5

12.6±0.2

0.033±0.003

55.6±5.0

6.57±0.50

0.067±0.007

1.376±0.109

0.311±0.011

1.162±0.10

0.556±0.010

0.340±0.020

L6

7.67±0.21

9.0±1.0

12.7±0.3

0.065±0.005

25.4±3.0

1.66±0.06

0.096±0.006

0.696±0.100

0.099±0.009

0.481±0.090

0.406±0.006

0.077±0.007

autumn

L1

7.54±0.10

5.6±0.1

15.4±0.4

0.057±0.007

47.5±2.0

4.49±0.49

0.244±0.034

2.874±0.074

0.317±0.017

2.603±0.060

1.088±0.088

0.172±0.002

L2

6.89±0.19

5.5±0.1

17.2±1.0

0.051±0.002

34.8±1.8

3.18±0.28

0.035±0.005

1.153±0.053

0.444±0.021

0.782±0.030

0.450±0.050

0.395±0.005

L3

7.11±0.10

4.1±0.1

14.6±0.6

0.012±0.001

83.2±3.2

10.77±1.74

0.539±0.039

5.005±0.202

0.609±0.041

3.882±0.080

3.517±0.017

1.007±0.007

L4

7.16±0.06

5.5±0.2

16.3±0.9

0.055±0.005

54.2±2.4

3.99±0.41

0.141±0.004

2.661±0.060

0.295±0.010

1.897±0.070

1.949±0.049

0.222±0.022

L5

7.02±0.02

4.5±0.1

14.1±0.1

0.015±0.005

47.8±2.0

5.25±0.79

0.071±0.001

1.454±0.054

0.369±0.009

1.051±0.051

0.554±0.054

0.611±0.011

L6

7.18±0.18

5.9±0.3

16.5±0.5

0.025±0.005

32.9±1.7

2.28±0.18

0.265±0.006

0.629±0.029

0.167±0.007

0.444±0.024

0.343±0.033

0.104±0.004

Note: pH, T, DO and DTP was measured in overlying water of sediments; TOC, TN, NH3-N, TP, IP, OP, HCl-P and NaOH-P was measured in sediments. Data are means±stand deviation.

11. Table 2: Same as above.

Reply: Formatting issues of table 3 have been fixed and standard deviation values have been shown in table 3.

Table3. The gcd-harboring bacteria richness and diversity in sediments of the Sancha Lake.

Season

Sampling site

Reads

Chao1

Shannon

coverage

OTUs

No. of

Phyla

No.of

Classes

No. of

Orders

No.of

Families

No. of

Genera

spring

L1

32585±150

234.55±33.05

3.365±0.040

0.9993±0.0004

226±6

6

9

13

19

22

L2

23251±120

156.18±21.04

2.777±0.067

0.9985±0.0012

150±4

5

9

12

15

23

L3

12834±60

98.60±19.04

1.539±0.030

0.9986±0.0007

85±2

2

4

7

11

16

L4

41828±200

185.27±29.04

2.622±0.064

0.9993±0.0005

158±4

1

3

5

10

17

L5

12151±60

140.69±18.09

3.167±0.071

0.9983±0.0002

130±3

3

5

7

10

13

L6

13357±65

322.14±36.64

3.397±0.075

0.9958±0.0017

291±7

3

6

9

17

32

autumn

L1

95695±50

163.05±23.04

2.943±0.040

0.9978±0.0014

154±4

5

8

12

16

22

L2

10880±54

143.96±20.05

1.259±0.014

0.9983±0.0004

121±3

2

4

5

5

15

L3

3199±10

43.00±1.00

1.307±0.016

0.9974±0.0000

29±0

1

3

5

6

10

L4

10852±54

80.20±3.04

1.939±0.029

0.9984±0.0006

62±2

1

3

5

7

10

L5

30857±150

139.38±17.04

2.432±0.039

0.9994±0.0001

118±3

2

3

5

8

12

L6

18415±92

157.53±22.04

2.273±0.027

0.9992±0.0005

152±3

5

8

13

21

23

Note: Data are means±stand deviation.

12: Figure 4: This figure is too busy to provide the information can you present in any other format if possible or a table instead of figure.

Reply: Figure 4 has been improved.

13: Conclusion: The authors have just mentioned the results in conclusion section. I don’t find any conclusion of the research conducted presented here and its impact on the scientific research field. I recommend authors to include the finding of the research and its conclusion here.

Reply: The findings and conclusions of our research have been included in the part “Conclusion”.

A total of 219,778 sequencesof high quality were obtained from sediment samples of Sancha Lake by DNA extraction, gene gcd sequences amplification with PCR and Illumina MiSeq platform sequencing. The gcd-harboring bacterial communities with confirmed classification information were composed of 6 phyla, 9 classes, 15 orders, 29 families and 46 genera. The gcd genes had a higher diversity. About 20% of the gcd-harboring bacterial communities cannot be identified. Therefore, the genetic gcd diversity in the sediments of Sancha Lake may actually be higher and new species may exist. From the center of the lake to the dam, and then to the end of the lake, the biodiversity index and abundance showed an upward trend. The gene gcd diversity index and abundance in the seriously polluted sampling sites were even lower. The bacterial diversity index of gene gcd in spring was higher than that in autumn. The abundance of these dominant gcd-harboring bacteria such as Rhizobium, Ensifer, Shinella and Sinorhizobium were higher in spring than in autumn, supposing that they have an important effect on eutrophication of the Sancha Lake. The major environmental factors affecting gcd gene diversity and gcd-harboring bacterial community composition are determined to be DO, TOC, TN, TP, HCl-P and DTP through Pearson correlation analysis and CCA analysis. The results indicated that GDH encoded by gcd genes catalyzed the oxidation of glucose into gluconic acid, resulting in insoluble phosphate transformation into soluble phosphate in the sediments. DTP concentration in overlying water increased, accelerate eutrophication of the Sancha Lake. gcd genes encoding GDH play an important role in lake eutrophication.

Reviewer 2 Report

The topic of GDH and it's contribution to eutrophication is very interesting and this case study shows some interesting results and conclusions.

Nevertheless several points needs to be improved:

1) English - even for non-native speaker it is obvious that there are some typing errors. Sometimes sentences are too long and their separation could help to easier understanding of sentence content. Maybe sometimes , is used instead of . ? I suggest to ask native speaker for language control or to use professional proof-reading.

2) In line 53-54 and 56-57 you mention that there are only few previous studies and research closely connected to your work. But you mention just one (reference 8). Other reference needs to be added to the text.

3) In chapter 3 results are well presented but discussion should be improved. Beside comparing your results to those few existing studies you mention in Introduction I'm missing mostly some discussion about possible impact of your results and connection with environmental problem you're dealing with. How could your results contribute to solution of eutrophication?

4) In figure 3 some colours aren't visible due to low relative abundance. Figure could be enlarged and bright colours could be used for lower abundant phyla (e.g. Verrucomicrobia is almost the same colour as one of unidentified) to improve it.

Author Response

Dear Editors and Reviewers:

Revised portion are marked using the "Highlight" function in the paper. The main corrections in the paper and the responds to the reviewer’s comments are as flowing:

Responds to the reviewer’s comments:

The topic of GDH and it's contribution to eutrophication is very interesting and this case study shows some interesting results and conclusions.

Nevertheless several points needs to be improved:

1) English - even for non-native speaker it is obvious that there are some typing errors. Sometimes sentences are too long and their separation could help to easier understanding of sentence content. Maybe sometimes , is used instead of . ? I suggest to ask native speaker for language control or to use professional proof-reading.

Reply: English has been checked and much improved to make it easier to understand.

2) In line 53-54 and 56-57 you mention that there are only few previous studies and research closely connected to your work. But you mention just one (reference 8). Other reference needs to be added to the text.

Reply: Other references have been added to the text.

3) In chapter 3 results are well presented but discussion should be improved. Beside comparing your results to those few existing studies you mention in Introduction I'm missing mostly some discussion about possible impact of your results and connection with environmental problem you're dealing with. How could your results contribute to solution of eutrophication?

Reply: Discussion has been improved and we add some discussion about possible impact of our results and connection with lake eutrophication.

4) In figure 3 some colours aren't visible due to low relative abundance. Figure could be enlarged and bright colours could be used for lower abundant phyla (e.g. Verrucomicrobia is almost the same colour as one of unidentified) to improve it.

Reply: Figure 3 has been improved.

Round 2

Reviewer 1 Report

The authors have addressed all the comments from previous comments.

Line 42. reference after eutrophication of water [1].

Line 121: remove "[T]"

Line 131: "The amplicon size was 330 bp [9] " should this reference be in between lines above                        line 131?

Table 2 needs formatitng 

Author Response

Dear editor

I have revised suggestion all according to the opinions.

thank you

best regards

 xu wenlai

Reviewer 2 Report

I see significant improvement of language and text.

I wasn't able to detect any improvement of Figure 3 (now Figure 4) which you reported.

Try to fix labels in Tables headers to not separate single words in two lines.

Author Response

(The authors gave the same response as above.)
